# Structural Health Monitoring of Composite Pipelines Utilizing Fiber Optic Sensors and an AI-Based Algorithm—A Comprehensive Numerical Study

**DOI:** 10.3390/s23083887

**Published:** 2023-04-11

**Authors:** Wael A. Altabey, Zhishen Wu, Mohammad Noori, Hamed Fathnejat

**Affiliations:** 1International Institute for Urban Systems Engineering (IIUSE), Southeast University, Nanjing 210096, China; wael.altabey@gmail.com; 2Department of Mechanical Engineering, Faculty of Engineering, Alexandria University, Alexandria 21544, Egypt; 3Department of Mechanical Engineering, California Polytechnic State University, San Luis Obispo, CA 93405, USA; 4School of Civil Engineering, University of Leeds, Leeds LS2 9JT, UK; 5Basque Center for Applied Mathematics, 48001 Bilbao, Spain; hamedfathnejat@gmail.com

**Keywords:** Fiber Bragg grating (FBG) sensory system, damage detection, structural health monitoring (SHM), deep learning, Convolutional Neural Network (CNN), composite pipelines

## Abstract

In this paper, a structural health monitoring (SHM) system is proposed to provide automatic early warning for detecting damage and its location in composite pipelines at an early stage. The study considers a basalt fiber reinforced polymer (BFRP) pipeline with an embedded Fiber Bragg grating (FBG) sensory system and first discusses the shortcomings and challenges with incorporating FBG sensors for accurate detection of damage information in pipelines. The novelty and the main focus of this study is, however, a proposed approach that relies on designing an integrated sensing-diagnostic SHM system that has the capability to detect damage in composite pipelines at an early stage via implementation of an artificial intelligence (AI)-based algorithm combining deep learning and other efficient machine learning methods using an Enhanced Convolutional Neural Network (ECNN) without retraining the model. The proposed architecture replaces the softmax layer by a k-Nearest Neighbor (k-NN) algorithm for inference. Finite element models are developed and calibrated by the results of pipe measurements under damage tests. The models are then used to assess the patterns of the strain distributions of the pipeline under internal pressure loading and under pressure changes due to bursts, and to find the relationship of strains at different locations axially and circumferentially. A prediction algorithm for pipe damage mechanisms using distributed strain patterns is also developed. The ECNN is designed and trained to identify the condition of pipe deterioration so the initiation of damage can be detected. The strain results from the current method and the available experimental results in the literature show excellent agreement. The average error between the ECNN data and FBG sensor data is 0.093%, thus confirming the reliability and accuracy of the proposed method. The proposed ECNN achieves high performance with 93.33% accuracy (P%), 91.18% regression rate (R%) and a 90.54% F1-score (F%).

## 1. Introduction

In order to meet safety and environmental regulations, it is vital to ensure the reliability of operating gas and oil pipelines. The failure of a large number of pipelines in several countries in recent years has led to big losses in human lives, destruction of residential and industrial buildings, and has caused complicated environmental hazards [1]. These disasters have resulted in increasing research activities dealing with the issues related to the failure prevention of pipelines.

Thodi et al. [2] pointed out that hydrocarbon leakage constitutes a serious impact on the safety of chemical equipment operations. Studies have shown that corrosion is the main cause of approximately 15% of leaks in chemical plants and accounts for 21% of gas pipelines failures. Corrosion patterns also account for 24.6% of pipeline leaks in processing plants. In addition, 40% of accidental hydrocarbon releases into the environment are related to corrosion. Papavinasam et al. [3] studied weight loss, linear polarization resistance (LPR), electrochemical impedance spectroscopy (EIS), the reliability of electrochemical noise (EN), and the performance of inhibitors used to monitor oil and gas pipelines using external hydrogen probes. Sinha [4] developed an ultrasonic sensing method for monitoring natural gas pipelines. He demonstrated how to use this technique to monitor various types of defects in pipelines and use a transducer in the pipeline to detect defects on the outside of the pipeline (for example, in a 0.5 mm groove on a 7 mm thick tube). Jawhar et al. [5] focused on the use of wireless sensor networks in petroleum pipelines and also for monitoring and protection of natural gas and water pipelines. Their proposed sensor network demonstrated that it could reduce installation and maintenance costs, reduce energy consumption, and increase the reliability and efficiency of pipeline operations. Ceravolo et al. [6] used the spectral entropies method for damage detection and localization of single and multiple points of damage in a buried steel pipeline by measuring the strain. They demonstrated that the wiener entropy or spectral flatness method is emerging as an efficient method for damage assessment.

It is critically important to provide effective and suitable SHM systems for structures, especially dangerous structures that would cause catastrophic loss of life if their failure occurred suddenly without warning. The main use of SHM systems is to diagnose the health and safety of structures over time through the collection of structural health datasets from sensors installed in the structure and analysis using assistant algorithms to predict the remaining life of the structure. Therefore, catastrophic accidents can be prevented before they happen by detecting the different stages of the damage over time prior to a potential failure [7,8,9,10]

Several studies have been reported on SHM methods for pipelines. Morison [11] introduced an SHM scheme for detecting internal corrosion in pipelines. Park et al. [12] proposed an SHM-based impedance method using piezoelectric materials to monitor damage in pipelines. They used a high-frequency excitation method to monitor the local structure area to detect the change in structural impedance associated with impending damage. Stoianov et al. [13] improved the SHM system proposed by Jawhar by increasing the spatial and temporal resolution for wireless sensor networks (WSN) based on real-time data monitoring. They demonstrated that WSNs could monitor large-diameter and large-scale water pipelines. Thien et al. [14] discussed the benefits and feasibility of applying an SHM system that relies on the deployment of macro-fiber composite (MFC) transducers for sensor arrays. Since the MFC patch is flexible, it can be permanently installed on the curved surface of the pipe body. To identify and locate damage in a pipeline, they studied Lamb wave-based MFC sensors to detect cracks and corrosion. Jin and Eydgahi [15] described a monitoring system for pipelines via a platform of sensor networks. They implemented their technique in pipeline systems for the distribution and transportation of oil, natural gas, water, and sewage. They discussed how their sensor network could detect, locate and quantify bursts, leaks, and other abnormalities in a pipeline system. Peairs et al. [16] utilized sensor nodes to monitor oil pipelines and studied the linear sensor placement problem to maximize their lifetime. Tapanes [17] presented current research on the impedance-based SHM technique at the Center for Intelligent Material Systems and Structures. They applied high-frequency excitations as the basic principle in their technique by using piezoelectric transducers to measure the structure’s impedance through the current and voltage monitoring. They found that the impedance methods have drawbacks and that methods based on this approach are expensive and not practical. Park and Inman [18] introduced a variety of SHM systems to monitor the integrity of pipelines while in operation and proved their approach could prevent catastrophic failures and reduce the costs of maintenance and inspection tasks.

Once Fiber-Optic (FO) Sensors appeared in the mid-1970s, they became highly valued by relevant research departments in various countries. The United States is the country with the earliest and the highest level of research on FO sensors, and its progress in military and civilian applications is very rapid. In terms of military applications, their research and development mainly concentrates on using FO sensors for underwater detection, FO sensors for aviation monitoring, FO gyroscopes, and FO sensors for nuclear radiation detection. In the mechanical and civil engineering fields, FO sensors are mainly used to monitor important parameters such as current, voltage, and temperature of the power system, and to monitor stress changes in bridges, pipelines, and also important buildings.

Numerous studies have been carried out on pipelines using FO sensors to estimate the effects of pipeline damage using both experiments and numerical prediction methods. Nikles and Briffod [19] introduced a technique to address the impact of blockages in hydrocarbon pipelines using a Fiber Bragg grating (FBG) sensing system which provides distributed sensing capabilities. Their proposed approach provided results that simulate the effect of pipeline blockage, which proved the validity of their introduced technique. Inaudi and Glisic [20] reported many important field application examples of fiber-optic (FO) sensing with the ability to measure temperature and strain at thousands of points with a single FO. Their approach demonstrated important applications for monitoring slender pipelines installed in oil wells and coiled tubing. Their approach could detect pipeline leaks and prevent the failure of pipelines installed in refineries and could also be used for detecting hot spots in high-power cables. Meinert et al. [21] proposed a method for detecting and preventing serious damage to pipelines mainly caused by interference from several noise sources. They showed their permanent monitoring semi-intelligent system could reduce the need for online inspection. Yan and Chyan [22] discussed the theoretical and numerical studies related to suppressing unfavorable FO nonlinearity and Stimulated Brillouin Scattering (SBS) using a statistical approach.

To develop a non-slippage FO, the bonding and point fixation methods were investigated experimentally and the critical effective sensing length for long-gauge fiber was studied [23]. A combination between an artificial neural network and a distributed FO vibration sensing (DOFVS) system based on long-distance fiber has been used to collect the responses in the vibration signal of soil around a pipeline [24,25,26,27]. In addition, the technology of the DOFVS system based on a phase-sensitive optical time-domain reflectometer (OTDR) was used to develop a water pipeline hydrostatic leak test [28] and fatigue damage identification for composite pipelines systems using electrical capacitance sensors [29].

According to the background of the literature presented, most of the conventional non-destructive testing (NDT) techniques are not applicable for current research work. It has been shown that the conventional methods do not provide the necessary information about either the current or future performance of composite pipelines systems and are not suitable for either new or old composite pipelines systems. In this case, FO sensors could be an effective alternative. In this study, a novel monitoring approach is developed to detect damage at an early stage in composite pipelines. The proposed approach relies on designing an integrated sensing-diagnostic SHM system with the capability to detect damage in composite pipelines at an early stage. This is achieved by implementing an artificial intelligence (AI)-based algorithm combining deep learning and other efficient machine learning methods using an Enhanced Convolutional Neural Network (ECNN) without retraining the model. The ultimate goal of the research is to provide pipeline operators with a continuous, real-time, active warning system for the detection of damages in composite pipelines using distributed strain sensors.

## 2. Methodology

The methodology used in this research aims for damage detection and safety assessment of composite pipeline structure via a highly effective and reliable approach, and then integrates this approach as a practical SHM which is reliable for old and new piping system operation.

The aim is to investigate the feasibility of developing and operating a SHM system for the early detection of damage occurring in composite pipelines. The flowchart of the proposed method is shown in Figure 1.

More specifically, this paper intends to:(1)Establish an understanding of the FBG sensor characteristics in a composite structure made of BFRP composite pipe.(2)Assess the effect of damage occurring in the BFRP composite pipeline on the displacement response from the dynamic signal for every FBG, including on dispersion, attenuation, and scattering.(3)Develop a sensory health monitoring platform for early detection of damage and damage classification of the composite pipeline.(4)Propose a novel AI-based SHM that integrates with the FBG sensors to provide damage classification in composite pipelines.(5)Suggest a hybrid approach to improve the accuracy of automatic identification of damage in pipelines by combining deep learning and machine learning in a new algorithm, known in this research as ECNN using a hybrid CNN + k-NN algorithm.(6)Verify the effectiveness and accuracy of the proposed AI algorithm based on the four indexes that include the true-positive rate (TPR), true-negative rate (TNR), false-positive rate (FPR), and false-negative rate (FNR), the accuracy rate (P), regression rate (R) and F1-score (F) can be determined for proposed AI algorithm.

## 3. A Finite Element Model (FEM) of a Composite Pipeline

### 3.1. Damage Model for Composite Structures

In recent years, there have been dramatic increases in composite material use in pipelines. This due to their unique advantages in mechanical properties, particularly specific strength and specific stiffness, as well as better fatigue resistance and damage tolerance capabilities.

The specific characteristic of the damage is that the three components of FRP composites (matrix, fiber–matrix interface, and fiber) do not fail at the same time. The most frequent type of damage is fatigue damage which is caused by cyclic loading effects. In this type of damage, the failure is caused by reductions in the residual strength in a part of a composite structure, depending on their type and size [30]. There are many forms of damage that can occur in a composite structure, such as matrix cracks, matrix-fiber debonding, fiber breakage, and delamination.

There are three regions in which failure mechanisms occur in composite structures: the matrix (I), matrix–fiber interface (II), and fiber (III). These are illustrated in Figure 2. Region I is distinguished by the coalescence of microcracks due to shear, which causes progressive damage over fatigue cycles. After that, the microcracks’ number and size increase with the number of cycles. Fatigue damage progresses to region II once the cracks approach the fibers along the matrix–fiber interface. The cracks grow in the direction of the fibers due to the normal stress and strain components and this also is responsible for delamination and then progression to region III, in which fiber breakage occurs, leading to fiber failure [31].

### 3.2. BFRP Composite Pipeline Properties

The FEM of a composite pipeline is established by ANSYS software, and then both free meshing and tetrahedron meshing are applied. The size of the minimum element and element type of the composite pipeline are selected as 0.005 m for element size and SHELL 181 for element type, respectively. The composite pipeline dimensions are 1 m long, the distribution of which is [−45, 45,−45, 45,−45]. The density of the model is $2.8\;g/{\mathrm{cm}}^{3}$. The internal radius ($r_{i}$) is 0.04 m. The external radius ($r_{o}$) is 0.043 m, and the length (L) is 1.0 m. The two ends of the pipe are both fixed as boundaries. Table 1 shows a detailed list of the mechanical parameters of the BFRP composite pipeline properties, where $E_{x}$, $E_{y}$, $E_{z}$ are the elastic modulus in the ‘x’, ‘y’ and ‘z’ directions, $G_{xy}$, $G_{xz}$, $G_{yz}$ are the shear modulus in the ‘xy’, ‘xz’ and ‘yz’ planes, and $\nu_{xy}$, $\nu_{xz}$, $\nu_{yz}$ are the Poisson’s ratio in the ‘xy’, ‘xz’ and ‘yz’ planes.

### 3.3. Damaged Pipeline System Modeling

The damage is modeled by reducing the stiffness of the pipeline. An enlarged view of damaged areas (marked in purple) in the pipeline is shown in Figure 3.

The FBG sensor is a type of non-destructive test used in situ on a structure to measure the generated vibration applied to the structure due to external excitation sources. When the vibration source is environmental loads (e.g., wind, traffic, human activity), then the structures are subjected to loads expressed as ambient oscillation or vibration.

The ambient excitation force applied in this work is numerically simulated by MATLAB (−0.4~0.4 N) as shown in Figure 4a. Its frequency range is first detected at approximately 1–400 Hz. Its spectrum is shown in Figure 4b. The ambient excitation force is perpendicular to the wall of the pipe. Three damage levels are introduced, i.e., location: D1: 0.42–0.48 m, D2: 0.52–0.58 m, and D3: 0.62–0.68 m. It is assumed that all the damage occurs towards 180–360° and occurs at the pipe internal surface. The damage cases D1 and D2 have the same damage range with different locations and D3 has a greater damage range than both D1 and D2 combined (D1 + D2) [32,33,34].

### 3.4. Modal Analysis of the Pipeline

The 1st, 2nd, 3rd, and 4th frequency mode shapes of the intact pipeline are listed in Figure 5 and Table 2 for different damage cases (D0–D3).

## 4. System Modeling and Simulation

### 4.1. Stress–Strain Analysis in a Stressed Thick-Walled Pipe

Figure 6 shows a stressed thick-walled pipe with the radius r. The solid line is an unstrained pipe and the dashed line is a strained pipe.

As shown in Figure 6, the strain components of the pipe can be expressed by:(1)$$\varepsilon_{H}=\frac{2\pi \left(r+u\right)-2\pi r}{2\pi r}=\frac{u}{r},\;\varepsilon_{L}=constant,\;\varepsilon_{r}=\frac{\left(\delta r+\delta u\right)-\delta r}{\delta r}=\frac{\delta u}{\delta r},$$
where $\varepsilon_{H}$ is the hoop strain, $\varepsilon_{L}$ is longitudinal strain, and $\varepsilon_{r}$ is radial strain.

By applying the stress–strain relation (tri-axial tresses) for a pipeline under hydrostatic internal pressure (P), the stress–strain relation can be written as:(2)$$\varepsilon_{L}=\sigma_{L}-\upsilon \left(\sigma_{H}+\sigma_{r}\right)=\sigma_{L}-\upsilon \left(\sigma_{H}-P\right)$$
(3)$$E\varepsilon_{H}=E\frac{u}{r}=\sigma_{H}-\upsilon \left(\sigma_{L}+\sigma_{r}\right)=\sigma_{H}-\upsilon \left(\sigma_{L}-P\right)$$
(4)$$E\varepsilon_{r}=E\frac{du}{dr}=\sigma_{r}-\upsilon \left(\sigma_{H}+\sigma_{L}\right)=-P-\upsilon \left(\sigma_{H}+\sigma_{L}\right)$$
where E is the elastic modulus, υ is Poisson’s ratio, $\sigma_{H}$ is hoop stress, $\sigma_{L}$ is longitudinal stress, and $\sigma_{r}$ is radial stress.

From the equilibrium of the pipe element shown in Figure 6, we can find the hoop stress differential equations by:(5)$$r\left(1-\upsilon \right)\frac{d\sigma_{H}}{dr}-r\left(1-\upsilon \right)\frac{dP}{dr}=0$$

By solving the above differential equations and applying the general boundary conditions of the pipe in the stress equation, the stress components in the pipe (see Figure 6) can be expressed by:(6)$$\sigma_{H}=\frac{Pr_{i}{}^{2}}{r_{o}{}^{2}-r_{i}{}^{2}}\left(1+\frac{r_{o}{}^{2}}{r^{2}}\right),$$
(7)$$\sigma_{r}=\frac{Pr_{i}{}^{2}}{r_{o}{}^{2}-r_{i}{}^{2}}\left(1-\frac{r_{o}{}^{2}}{r^{2}}\right),$$
(8)$$\sigma_{L}=\frac{Pr_{i}{}^{2}}{r_{o}{}^{2}-r_{i}{}^{2}}\;for\;tube,\;\;\sigma_{L}=0\;for\;pipe$$

The pipe deformation (δ=u) in Figure 6 can calculated by Equation (9):(9)$$\delta =\frac{\left(1-\upsilon \right)}{E}\left(\frac{Pr_{i}{}^{2}}{r_{o}{}^{2}-r_{i}{}^{2}}\right)r+\frac{\left(1+\upsilon \right)}{E}\left(\frac{Pr_{i}{}^{2}r_{o}{}^{2}}{r\left(r_{o}{}^{2}-r_{i}{}^{2}\right)}\right),$$

### 4.2. Structural Health Monitoring (SHM)

SHM is a new field of research and development that emanated from the study of smart materials and structures. SHM has attracted considerable attention in recent years for assessment in infrastructure and aerospace vehicle applications. The goal of SHM is to develop automated systems that can provide continuous monitoring, inspection, and detection of damage to structures, thereby minimizing the need for human labor. A typical SHM system embraces three main sub-systems as shown in Figure 7: (i) a sensory system, (ii) a data processing system (including data acquisition, transmission, and storage), and (iii) a health assessment system (including diagnostic algorithms and information management). At a higher level, an SHM system can include a fourth sub-system with repair capabilities [35,36,37].

### 4.3. Fiber-Optic Sensors

Sensing technology is one of the fastest-growing forms of technology in the world today. The new sensors not only pursue high precision, large range, high reliability, low power consumption, and miniaturization but are also developing towards integration, multiple functions, intelligence, and networking to meet the needs of various fields such as industry, agriculture, national defense, and scientific research.

#### 4.3.1. Common Types of Fiber-Optic Sensors 

Selecting the correct FO sensor types to monitor the excitation forces applied to the structure is important for detection. Before examining the details of the working principle of FO, a concise review of the common types of FO sensors is provided. Among the common types of FO sensors, point sensors, and distributed sensors are of interest in the pipeline monitoring field. Figure 8 presents a classification of FO sensors based on guiding optical principles. These include distributed sensors for distributed sensing in large structures, short-gauge sensors for point-sensing in homogeneous materials, such as steel, and long-gauge sensors for point-sensing in heterogeneous materials, such as concrete.

#### 4.3.2. Fiber Bragg Grating (FBG) Sensors Description

The FBG system consists of an article interrogator that projects infrared light into the core of an optical fiber. As a white color, broadband light travels along the fiber, passing through grating segments, which are also known as FBG and consist of a series of article filters. These grid segments can filter out certain wavelengths or colors while letting others through.

This happens by periodically changing the refractive index of the fiber, which determines which wavelengths can pass through and which are reflected. External factors such as heat and vibration cause the reflected light to change wavelength. These variations can then be translated into physical engineering units such as temperature and stress. A type of FBG detection technology is shown in Figure 9.

#### 4.3.3. FBG Sensors Working Principle

The distributed FBG detection method uses FBG as a sensor to detect vibration signals along the pipeline. The optical cable is affected if ambient events such as human activities or mechanical operations occur close to sensing area. Such events generate vibration signals and cause the strain in the optical cable, resulting in the phase of light in the optical cable. Hence, the polarization state changes, and the system recognizes and registers the detected changes (see Figure 10). As illustrated in Figure 10, the Fiber Bragg grating (FBG) sensor system is integrated with a broadband light source, FBGs, a wavelength interrogator, and system software.

When the broadband light is projecting at an FBG, reflection at the FBG occurs. Some light with wavelengths that satisfy Bragg condition of Equation (10) is reflected, and the remaining light passes the grating:(10)$$\lambda_{B}=2n_{e}A$$
where $\lambda_{B}$ is the Bragg wavelength, $n_{e}$ is the effective refractive index, and A is the grating period. When strain is produced in an FBG, a proportional shift of the Bragg wavelength is expected to occur. To determine the strain easily, the change in wavelength is analyzed. This is the principle used detect the grating period change in FBG sensors due to stress variation; therefore, the stresses can be measured without noise influence and light intensity disturbance. The wavelength shift is proportional to the strain, and absolute strain can be measured by Equation (11):(11)$$\frac{\Delta \lambda_{B}}{\lambda_{B}}=\left\{1-\frac{n_{e}{}^{2}}{2}\left[P_{12}-\upsilon \left(P_{11}+P_{12}\right)\right]\right\}\varepsilon_{B}$$
where $P_{ij}$ are the silica photo-elastic tensor components, $\varepsilon_{B}$ is the strain of the fiber grating and υ is the Poisson’s ratio.

#### 4.3.4. Pipeline Monitoring Based FBG Sensor Technology

The FBG sensors that have been developed can be used for detecting damage in the pipelines. These include FBG-based pressure sensors for finding the point of leakage. The mechanisms of these sensors depend on, when a leak initially occurs, the pressure drop in a pipeline in either direction of the leakage point. Determining the point of leakage depends on the time taken for the wave to reach the FBG-based pressure sensors. The advantages and possible challenges of the FBG sensor system are as follows:

FBG sensors for pipelines can not only detect damage but also provide early warning of hazardous pipeline events that occur along the pipeline. They have features not found in traditional sensing technology. The key FBG sensor technologies are described as follows: The ability to measure the temperature and strain of thousands of points with a single fiber is very important for the inspection of slender structures (such as pipelines, pipelines, oil wells, and coiled tubing);

(1)Reliable monitoring and locating of small and slow leakages of gas, oil, heating, etc.;(2)The ability to identify man-made pipeline damage and provide real-time alarm and positioning with a very high accuracy rate and very low false alarm rate;(3)The system can provide an intelligent alarm function based on the geographic information system (GIS) platform for monitoring requirements;(4)Continuous distributed measurement along the line without blind spots, with self-diagnosis function, real-time detection and localization of damage by the detection sensor system;(5)One optical cable can perform temperature strain detection and communicate at the same time;(6)High-temperature and low-temperature resistance, long-distance testing, centralized ground signal processing;(7)Passive, intrinsically explosion-proof, especially suitable for use in flammable and explosive environments;(8)Good anti-corrosion and anti-interference performance;(9)Long-term stability and measurement accuracy which are not affected by the loss of transmission fiber.

#### 4.3.5. Design Theory of FBG Strain Sensor Array

The sensor cannot gauge the hoop strain at a certain point until it is closely adhered to the outer wall surface of the pipeline. In this work, due to the advantages of FBG distributed sensing, a single sensor with a multi-point monitoring system is used to monitor external strain changes. The FBG distributed sensor is installed on the outer wall surface of the pipeline with arbitrary orientation (φ) as shown in Figure 11a. The relative parameters of the packaging FBG parameters are selected as shown in Table 3.

From Figure 11b, the length of the FBG sensor ($L_{B}$) can be approximated as:(12)$$L_{B}=\pi DN_{t}$$
where $N_{t}$ is the total number of the sensor coils, and D is the outer diameter of the sensor coils (see Figure 11b). $N_{t}$ can be calculated by the formula:(13)$$N_{t}=\frac{L-D_{B}}{q_{B}},\;D=2r_{o}+D_{B}$$
where $D_{B}$ is the sensor wire diameter, and $q_{B}$ is the pitch between the grating. Finally the length of FBG sensor can be estimated as:(14)$$L_{B}=\pi \left[\left(2r_{o}+D_{B}\right)\frac{L-D_{B}}{q_{B}}\right]$$

The proposed sensor sensitivity can be determined by assuming that the relationship between the pipe hoop strain $\varepsilon_{H}$ and the fiber grating strain $\varepsilon_{B}$ is based on the fact that the hoop deformations of the sensor and pipe structures are the same, i.e., the effect of sensor stiffness on the pipe is ignored. Moreover, the loss of strain due to the sensor fixation type is not considered. The pipe hoop strain $\varepsilon_{H}$ and the fiber grating strain $\varepsilon_{B}$ can be expressed as follows:(15)$$\varepsilon_{B}=\frac{∆L_{B}}{L_{B}}=\frac{M}{E_{B}A_{B}}$$
where M is the internal force through the sensor due to a certain change in the pipe interior, and the hoop strain at the outer radius of the pipe ($r=r_{o}$) is given as:(16)$$\varepsilon_{H}=\frac{\sigma_{H}}{E}=\frac{2Pr_{i}{}^{2}}{E\left(r_{o}{}^{2}-r_{i}{}^{2}\right)},$$

And From Equation (12): (17)$$r_{o}=\frac{q_{B}}{2\pi }\left(\frac{L_{B}}{L}\right),$$

The relationship between the pipe hoop strain $\varepsilon_{H}$ and the fiber grating strain $\varepsilon_{B}$, taking into consideration the angle of the sensor, φ as shown in Figure 11b, can be expressed as follows:(18)$$r_{o}=\frac{q_{B}}{2\pi }\left(\frac{L_{B}}{L\frac{\varepsilon_{B}}{\varepsilon_{H}}=\cos\varphi =\frac{ME}{2PE_{B}A_{B}}\left({\left(\frac{q_{B}}{2\pi r_{i}}\left(\frac{L_{B}}{L}\right)\right)}^{2}-1\right)}\right),$$

$K_{B}$ is the strain sensitivity coefficient of the FBG strain sensor array and is given as:(19)$$K_{B}=\frac{L_{B}}{L}=0.86\pi ,$$

The relationship between the strain and the wavelength of the grating in this band can be approximated as:(20)$$\varepsilon_{B}=\frac{\Delta \lambda_{B}}{0.86\pi },$$

Therefore, the relationships between the center wavelength of the grating, the hoop strain $\varepsilon_{H}$, and internal pressure P at the outer radius of the pipe ($r=r_{o}$) are determined by the Formulas (14), (17) and (18):(21)$$\varepsilon_{H}=\frac{\Delta \lambda_{B}}{0.86\pi \cos\varphi },$$
(22)$$P=\frac{\Delta \lambda_{B}E\left(r_{o}{}^{2}-r_{i}{}^{2}\right)}{1.72\pi r_{i}{}^{2}\cos\varphi }$$

Figure 12 shows the relationship between the wavelength of the different gratings of the sensor, the hoop strain $\varepsilon_{H}$, and the internal pressure P at the outer radius of the pipe ($r=r_{o}$).

As shown in Figure 12 the linear relation between pressure wavelength, and hoop strain with sensor correlation factor reaches 0.9677, which indicates there is little difference between sensor gratings. From Figure 12 it can be concluded that the proposed sensor array has stable performance and is sensitive and suitable for monitoring the pipeline subject of the current study.

### 4.4. The BFRP Pipeline Damage Identification Model

In the Section 4.3.5, the sensitivity, stability, and linearity of the proposed FBG sensor network were verified for monitoring the pipeline, and the sensor response (pipeline displacement) was studied for the intact and damaged pipeline system. However, the various levels of damage are difficult to identify due to the inherent complexity in 3D modeling. Therefore, in order to identify the damage, an efficient method must be used to extract the FBG sensor response features. In damage identification problems, a hybrid CNN + k-NN (ECNN) can be used to test the hypothesis in the presented research. It can record better results than a traditional network CNN + softmax (TCNN). To evaluate this, a trained CNN classification response of the neural codes learned from the same CNN are compared, but a k-NN classifier is applied to the output of the last hidden layer. Three configurations are evaluated to improve the accuracy of the CNN algorithm after training with standard backpropagation (BP) in the inference stage (see Figure 13):(1)Use of the CNN softmax layer, which is known in this research as TCNN.(2)Using the last hidden CNN layer (before the softmax layer) to obtain the neural codes that are passed to a k-NN which compares them to the prototypes of the training set using the Euclidean distance in order to obtain the most likely class, known in this research as ECNN.(3)Using the k-NN directly on the raw data without any representation learning.

#### 4.4.1. A k-Nearest Neighbor (k-NN) Algorithm

Because of its ease of implementation and high efficiency, the k-NN algorithm is considered one of the most successful in past SHM applications. The method of k-NN application is based on searching for k points in a reference database for the closest points to the measured data according to a function that represents the distance between them that represents the optimal solution of minimum distance values of k. 

As the target of this new algorithm, the sensor displacement datasets and TFS maps before and after damage are used as training features for damage detection in the composite pipe k-NN damage prediction datasets with the following steps:(1)Integer k should be assigned first.(2)According to dataset distribution, the optimized k is determined so that we can iterate the integer number to be the best k that results in the highest accuracy.(3)The damage datasets pass through min-max normalization and z-score standardizations, as shown in Equations (23) and (24).(4)This is because when a pair of features is inputted, the k-NN searches the nearest k-pair of features using Euclidean distance on the same scale.
(23)$$Min-Max\;normalization\;\left(X\right)=\frac{\left(X-\min\left(X\right)\right)}{\left(\max\left(X\right)-\min\left(X\right)\right)}$$
(24)$$z-score\;standardization\;\left(X\right)=\frac{\left(X-\;\mathrm{main}\;\left(X\right)\right)}{\mathrm{StdDev}\;\left(X\right)\;}$$

The performance of k-NN results depends on the effectiveness of the method to measure the distance between the model datasets feature and new test inputs. The Euclidean method is the optimal method usually used for distance calculations between test and trained data. We measure the distance along a straight line from point $\left(x_{1},y_{1}\right)$ to point $\left(x_{2},y_{2}\right)$.
(25)$$Euclidean\;Distance=\sqrt{\overset{n}{\underset{i=1}{\sum }}{\left(x_{i}-y_{i}\right)}^{2}}$$

Figure 14 shows the k-NN algorithm. For selecting the optimum neighbors number (k), a genetic optimization method is used. 

Figure 15 shows the flowchart of the proposed genetic algorithm (GA) for selecting the optimum k. At too small values of k (k<10), the prediction error of the sensor displacement is very high, and the error decreases by increasing k from 10 to 100. The least error of prediction occurs at k=100 and at (k>100), the error increases again. Table 4 presents the k-NN internal parameters of the current study.

#### 4.4.2. The Convolutional Neural Network (CNN) Modeling

The connection architecture of the proposed CNN layer is presented in Figure 16. As shown in Figure 16, the CNN structure generally has a multi-layer architecture, including convolution, pooling, activation, and full connection layers. The input of the network is TFS images of the pipeline before and after damage. The significant local features are then extracted via the convolutional and pooling layers. Finally, the damage identification is outputted with the full connection layer. 

The fully connected layers in the CNN architecture are located as shown in Figure 17 between the input (features) and the output (target prediction) layer for the extraction of higher-level features through the training process. 

Using weights w and biases b, the CNN is trained for identifying more useful nonlinear information. In the proposed CNN, this process is defined as:(26)$$x_{j}^{l}=f\left(\underset{i}{\sum }x_{i}^{l-1}w_{ij}^{l}+b_{j}^{l}\right)$$
where $x_{j}^{l}$ is the *i*^th^ output map in layer l; $x_{i}^{l-1}$ is the *i*^th^ output map in layer l-1; $w_{ij}^{l}$ is the weight; $b_{j}^{l}$ is the bias; and f(·) is a nonlinear function that is applied component-wise.

One important stage of the neural network is the activation function that gives it the required nonlinearity, where a neural network becomes simple linear without it (essentially a probability distribution).

Softmax is used for multi-classification problems, where it normalizes the neural network’s output to fit between “0” and “1”. It is applied to represent the certainty “probability” in the network output. The expression of the softmax activation function is given as Equation (27):(27)$$\sigma {\left(\vec{z}\right)}_{i}=\frac{e^{z_{i}}}{\sum_{j=1}^{K}e^{z_{i}}}$$
where $\vec{z}$ is the input vector; $z_{i}$ values are the elements of the input vector; $e^{z_{i}}$ is the standard exponential function applied to each element of the input vector, and K is the number of classes in the multi-class classifier.

The pooling layer is usually arranged between sequential convolution layers. It is used to reduce the feature maps’ locative size. This is also called undersampling, by which network overfitting can be controlled. The operations that can be used for undersampling are maximum pooling and average pooling. It can be expressed as the average pooling feature of the pooling layer in Equation (28), assuming the pooling size is *c*, *j*th is the region, and *l*th is the number of pooling layers:(28)$$x_{j}^{l}=f\left(B_{j}^{l}mean\left(x_{j}^{l-1}\right)+b_{j}^{l}\right)$$
where $B_{j}^{l}$ is multiplicative; and mean(.) is the average operation. The convolution and pooling layers work together to detect the local connections, merge similar features and remove unnecessary irrelevant details.

## 5. Results and Discussion

### 5.1. The Displacement Response Identification

Figure 18 shows the displacement of ambient excitation measured from the FBG sensor for the undamaged pipe (UDP) model D0 and damaged pipe (DP) models D1, D2, and D3, respectively. The boundaries are fixed for both two ends of the pipeline. At a distance of 0.2 m from the pipeline right support is the position at which the ambient forces are applied. The various levels of damage along the pipeline are considered.

Figure 19 shows the time–frequency spectrogram (TFS) for the undamaged pipe (UDP) model D0 and damaged pipe (DP) models D1, D2, and D3, respectively. The TFS in this paper are extracted from the direct wave packet before and after pipe damage, and the frequency of the Gaussian wavelet transform is set from 50 Hz to 300 Hz with an interval of 1 Hz. As shown in Figure 19, the TFS images have a size of 201×213. In Figure 19a–d, the magnitude of the direct wave packet is increased because the reflected wave due to pipe damage increases when the damage changes.

### 5.2. Experimental Validation of the Proposed Method

In this section, we present some experimental verifications of the proposed approach by applying the present technique to an experimental dataset adapted from Wang et al. [38]. They used an optical fiber sensing system with a configuration of fiber Bragg grating (FBG) to extract the damage behavior in CFRP composite pipes. The ultimate goal of this section is to extract the experimental datasets of Wang et al. [38] and compare them with the numerical datasets presented in this work that are derived from the sensory network of an FBG series installed on the outer surface of pipes. The aim is to prove the effectiveness and feasibility of the proposed sensing technique.

The vibration experiments were performed on the cantilever CFRP pipes shown in Figure 20. Excitation frequencies of 10 Hz acted on the specimens with a loading of about 2000 s. The dimensions of the specimens are 12 mm, 15 mm, and 100 mm for the inner diameter, outer diameter, and length, respectively. Three series of FBGs were installed on the outer surface of the pipe as follows: the first two series were composed of three FBGs, and the last one had four FBGs.

The vibration strains values at x=0.75 m and at the adjacent FBGs points were extracted numerically from the FEM, and also from the experimental FBGs and compared. Figure 21 presents the distribution of the vibration strains for both the computed and measured values over the time.

### 5.3. Hybrid CNN + k-NN (ECNN) Architecture as a Surrogate Model

In the last layer of the CNN, the learned feature maps are flattened into one vector by a fully connected layer and the expected output is extracted. In this paper, the displacement response evaluation problem for undamaged and damaged pipes is a regression problem. Therefore, a softmax activation function or k-NN algorithm is adopted in the full connection layer, by which a vector value denoting the displacement response is outputted. As discussed in Section 4.4, the proposed model (ECNN) architecture includes two convolutional layers, two sub-sampling layers, and a fully connected layer. Each layer tunes the parameters and the corresponding weights. The overall process of the proposed ECNN is described in Figure 22.

The prediction of damaged pipeline displacement can be described through the following steps in Figure 23. If the ambient excitation σ(x,t) and the initial condition are given, the proposed ECNN can be used to predict the displacement response of the pipeline (Figure 24).

#### The Displacement Response Prediction Based on the ECNN

As shown in Figure 4, an ambient excitation load σ(x,t) in a composite pipeline is used to obtain the training data. The trained model is used to predict the displacement response of the damaged pipeline under that excitation load. Figure 25 shows the training data. As shown in the figure, the proposed deep learning model can predict the displacement response of the composite pipeline under ambient excitation load. The results show an excellent agreement between the FBG sensor data and ECNN data with an average error of 0.093%.

Table 5 labels all the cases regarding different damage cases (D0–D3). Table 6 shows the comparison between the three AI configurations for damage identification results for four labels with respect to true positive rate (TPR), true negative rate (TNR), false positive rate (FPR) and false negative rate (FNR). 

In general, for all indexes (TPR, TNR, FPR, and FNR), using k-NN over the input datasets resulted in a lower average accuracy than the TCNN configuration. The hybrid approach ECNN achieved better results than the TCNN or k-NN. As a general conclusion, the proposed ECNN approach consistently outperformed the TCNN and k-NN for all indexes.

To estimate the proposed ECNN performance in damage identification of BFRP composite pipelines, three indicators were calculated during the training process. The accuracy rate (P%), regression rate (R%), and F1-score (F%) are based the indexes (TPR, TNR, FPR, and FNR) and are calculated as follows:(29)$$P\%=\frac{\mathrm{TPR}}{\mathrm{TPR}+\mathrm{FPR}}$$
(30)$$R\%=\frac{\mathrm{TPR}}{\mathrm{TPR}+\mathrm{FNR}}$$
(31)$$F\%=\frac{2\;\mathrm{TPR}}{2\;\mathrm{TPR}+\mathrm{FNR}+\mathrm{FPR}}$$

Figure 26 shows the calculated P%, R%, and F% for 800 testing datasets divided into ten groups of displacement results (i.e., D1, D2,….., and D10) after 350 iterations to verify the effectiveness of the proposed ECNN. They are presented as bar plots with different colors, and the overall performances of selected displacement is indicated with a dashed line.

As shown in Figure 26, the overall performance values were 93.33%, 91.18%, and 90.54%, for P%, R%, and F%, respectively. These results confirm that the proposed method can automatically identify the damage in composite pipelines with satisfactory performance regardless of the corresponding capacitance data noise background and conditions.

As seen in Figure 26, based on the results of testing with 800 datasets, the proposed method applying ECNN to the identification of different extents of damage in pipelines is promising and may be suitable for other composite structures.

## 6. Conclusions

Based on the response of the FBG sensor system of the damaged composite pipeline, a novel AI-based algorithm is proposed that combines deep learning and machine learning utilizing ECNN without modifying the training stage for the displacement response prediction of the composite pipeline. The proposed architecture replaces the softmax layer in TCNN with a k-NN algorithm for inference. The proposed ECNN model was divided into two networks: a response and frequency network. The frequency network converges to the shape mode of the pipeline, and the response network is a feedback network for predicting the displacement response a long pipeline. A composite pipeline made of the BFRP was analyzed using an FEM to simulate the displacement of the pipeline. Three damage levels were introduced to validate the effectiveness of the proposed approach. The training data were generated by the FEM. From the results, it can be concluded that the proposed AI-based model can effectively predict the displacement response of composite pipelines, and works much faster than in terms of computational time of the traditional FEM. The results show that replacing softmax with k-NN significantly outperforms the TCNN architecture in terms of accuracy. The proposed method achieved satisfactory performance, with the values of P%, R%, and F% being 93.33%, 91.18% and 90.54, respectively. Therefore, the proposed method has special advantages for solving practical engineering problems.

## Figures and Tables

**Figure 1 sensors-23-03887-f001:**
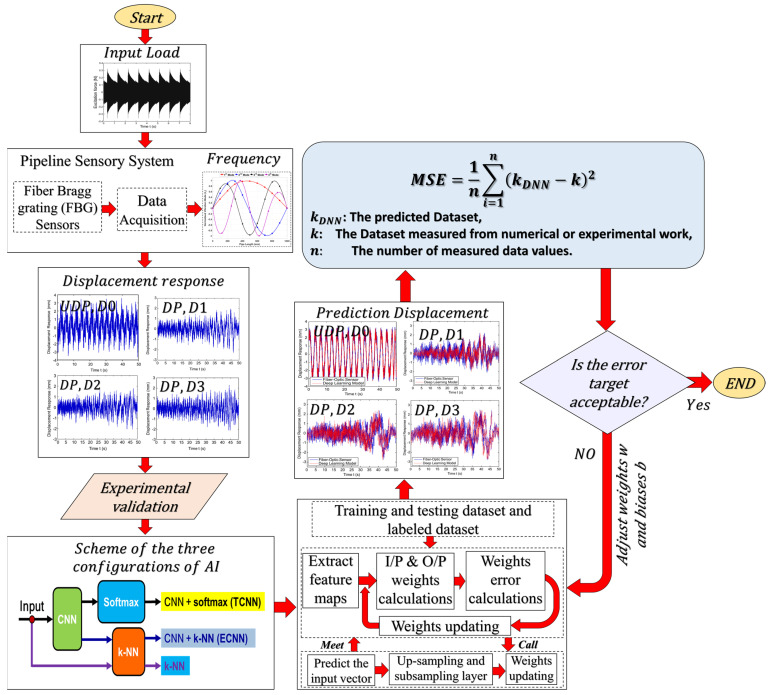
Methodology flowchart.

**Figure 2 sensors-23-03887-f002:**
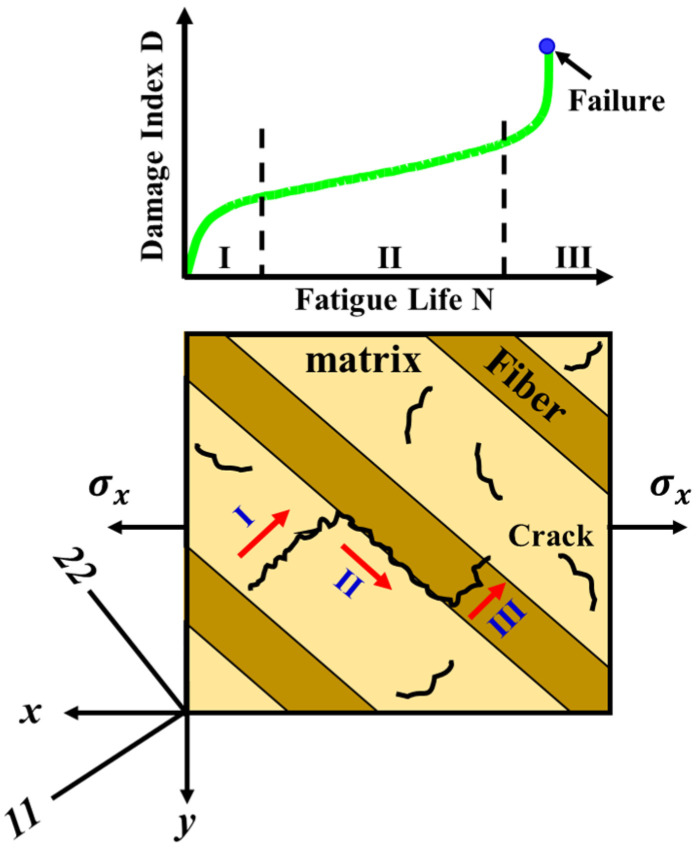
Three Regions of Cracking Mechanisms in Unidirectional Composites.

**Figure 3 sensors-23-03887-f003:**
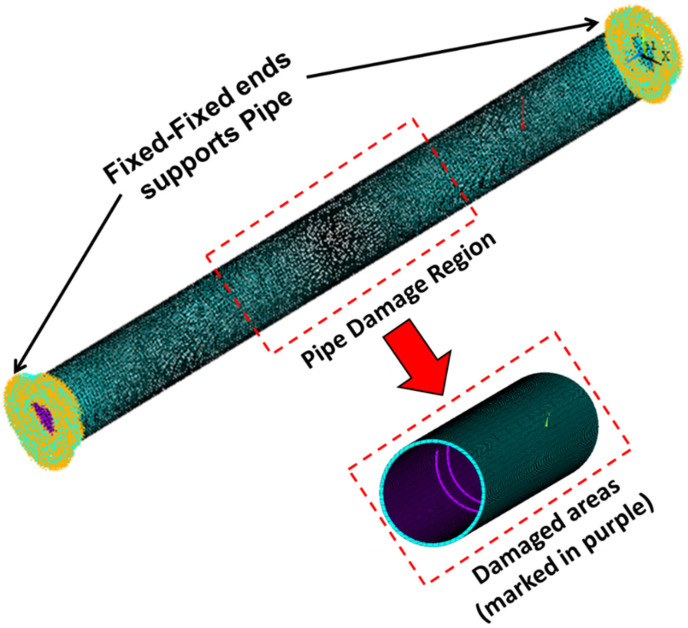
Fixed ends supporting pipe.

**Figure 4 sensors-23-03887-f004:**
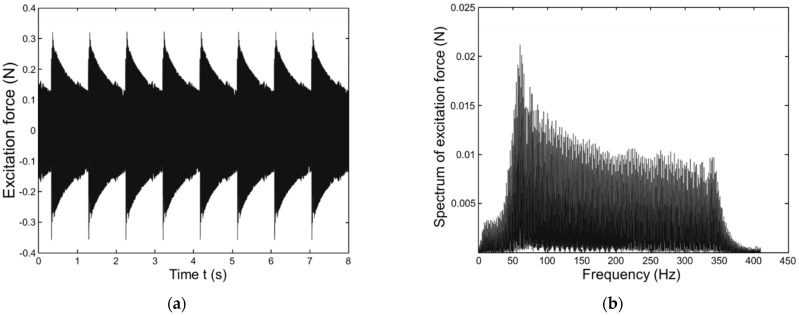
Ambient excitation force. (**a**) Time domain. (**b**) Spectrum.

**Figure 5 sensors-23-03887-f005:**
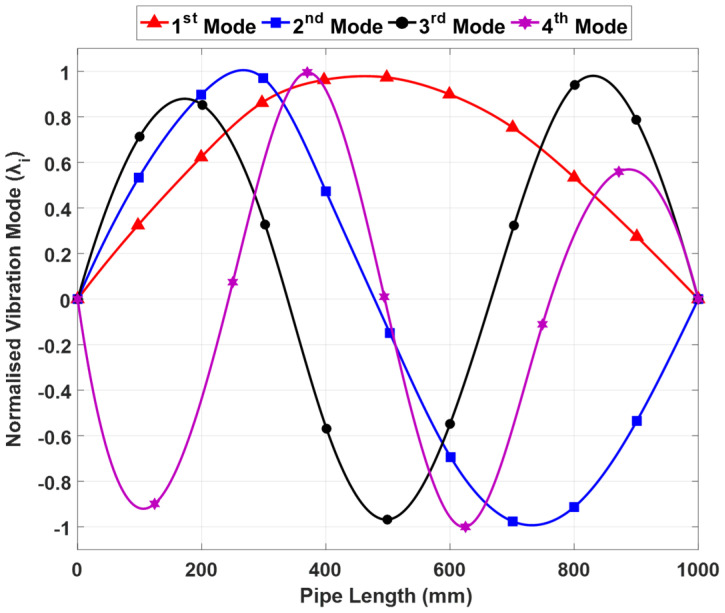
The first four mode shapes of an intact pipeline.

**Figure 6 sensors-23-03887-f006:**
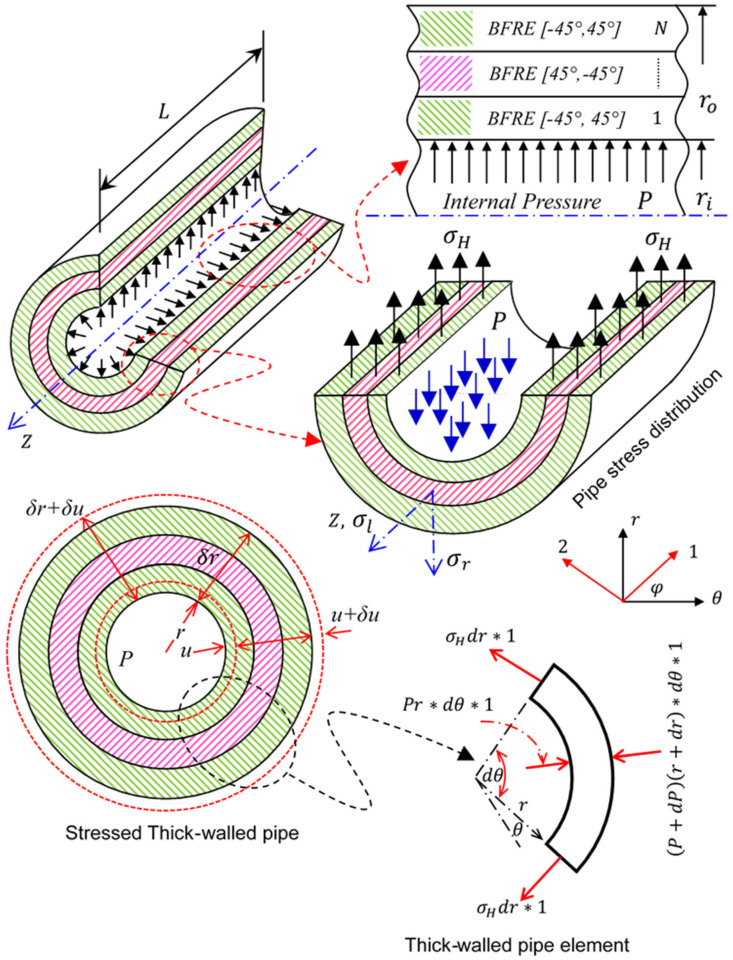
Stressed thick-walled pipe.

**Figure 7 sensors-23-03887-f007:**
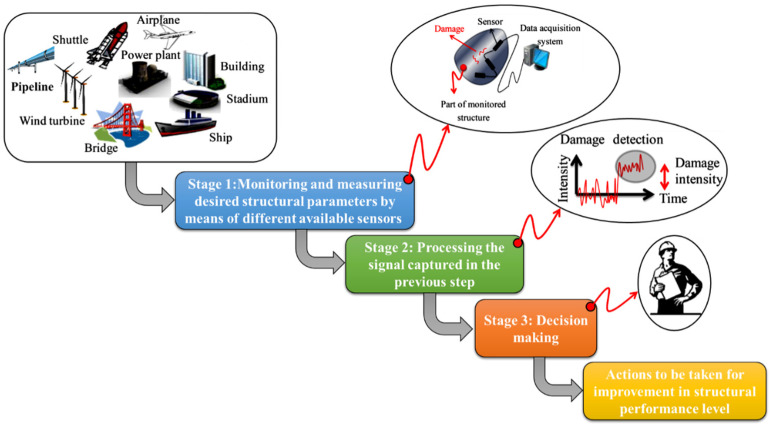
SHM system main stages.

**Figure 8 sensors-23-03887-f008:**
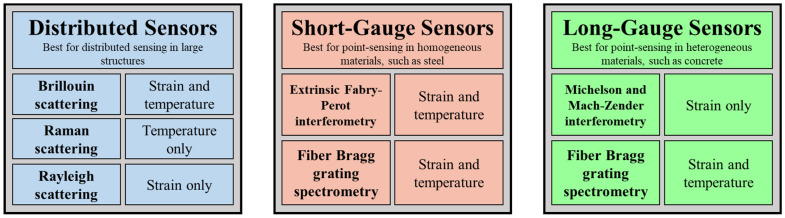
Schematic diagram of classification of FO sensor based on application type.

**Figure 9 sensors-23-03887-f009:**
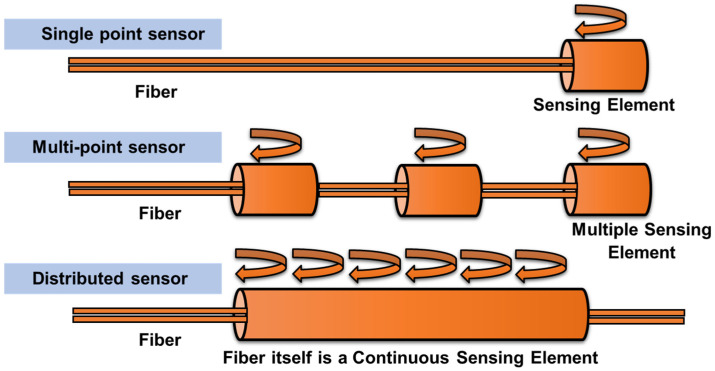
Schematic diagram of three FBG Sensors.

**Figure 10 sensors-23-03887-f010:**
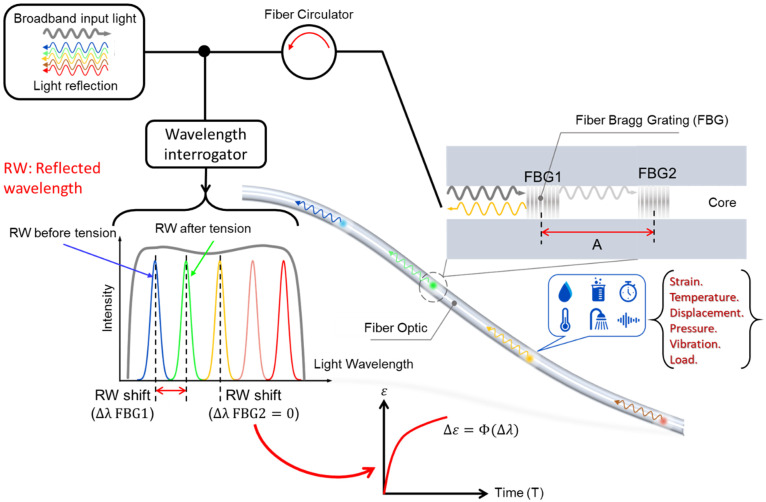
Schematic of the working principle of Fiber Bragg grating (FBG) sensors.

**Figure 11 sensors-23-03887-f011:**
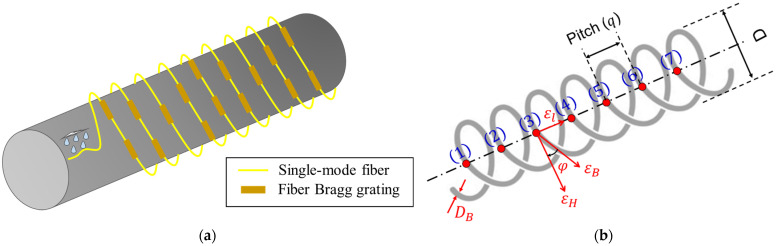
The Theory of FBG Strain Sensor Array (**a**) FBG distributed sensors installed in the pipeline (**b**) The relationship between the pipe hoop strain $\varepsilon_{H}$ and the fiber grating strain $\varepsilon_{B}$.

**Figure 12 sensors-23-03887-f012:**
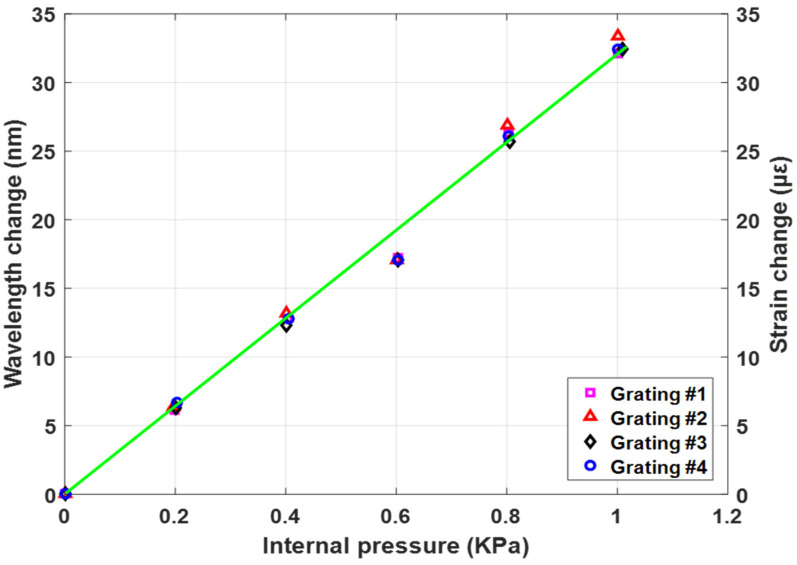
The relationship between internal pressure of pipe and wavelength of the grating and the hoop strain.

**Figure 13 sensors-23-03887-f013:**
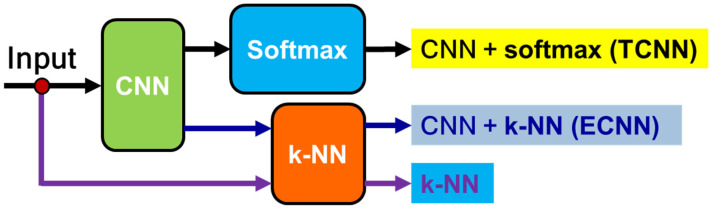
Scheme of the three configurations for improving accuracy.

**Figure 14 sensors-23-03887-f014:**
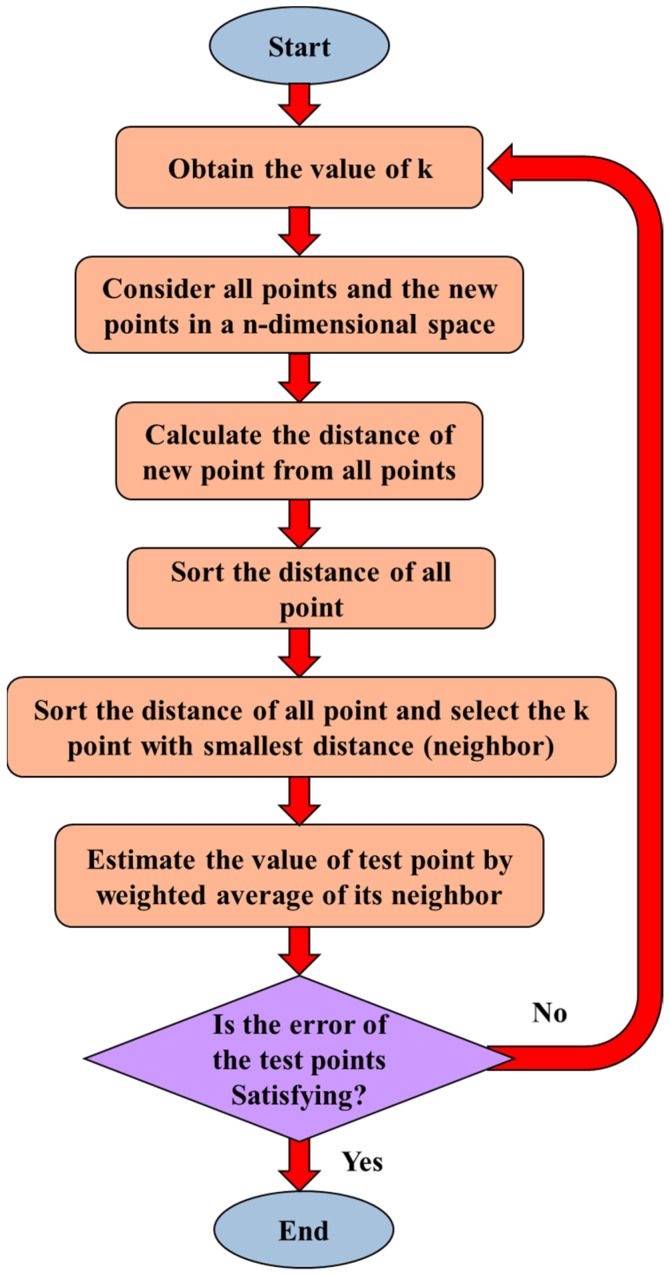
Flowchart of the k-NN algorithm.

**Figure 15 sensors-23-03887-f015:**
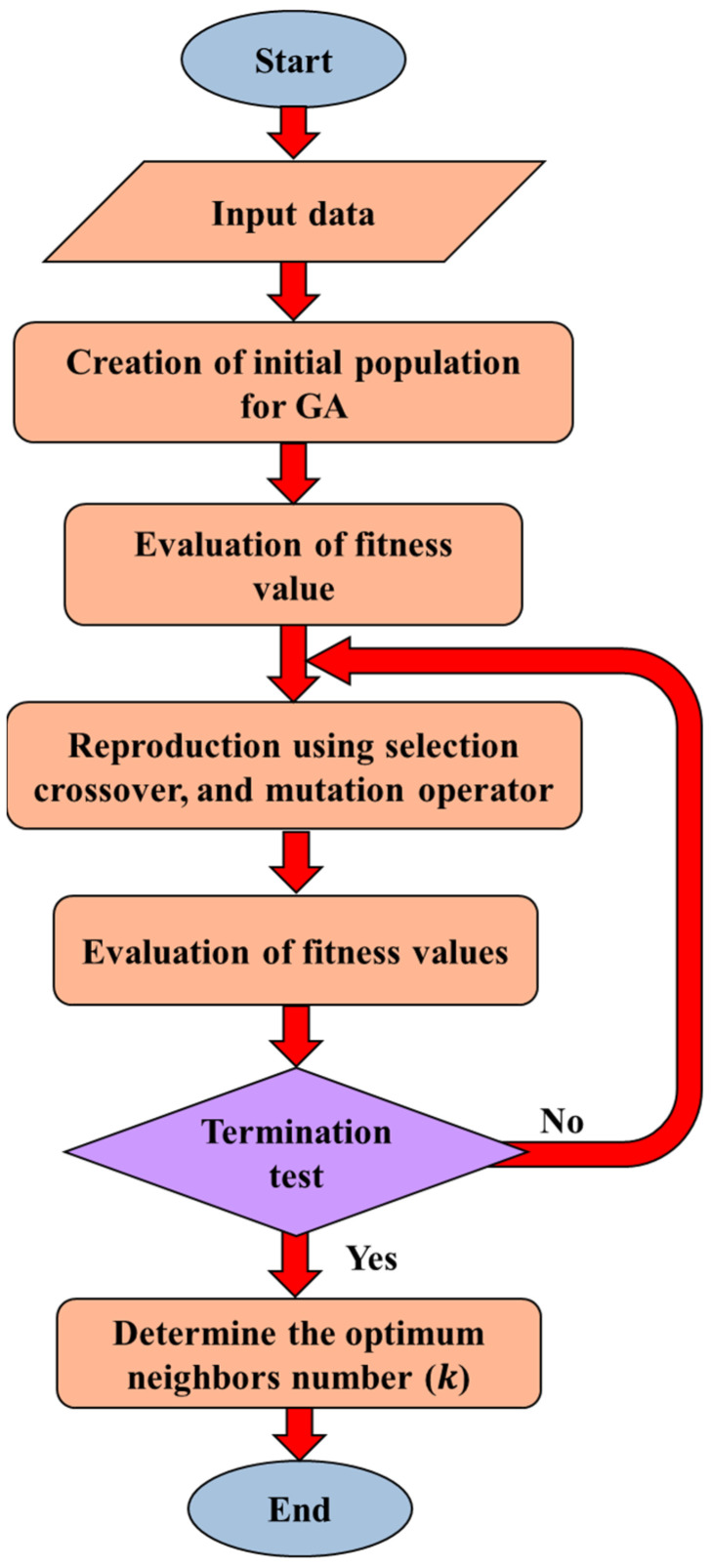
Flowchart of the proposed genetic algorithm (GA) for selecting the optimum *k*.

**Figure 16 sensors-23-03887-f016:**
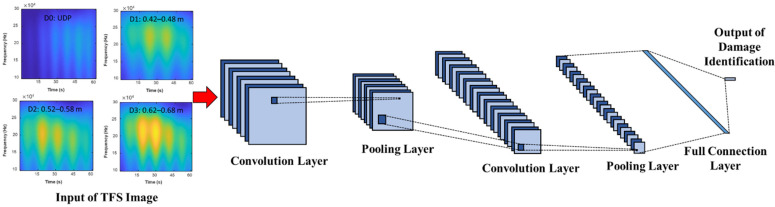
The architecture of a typical CNN.

**Figure 17 sensors-23-03887-f017:**
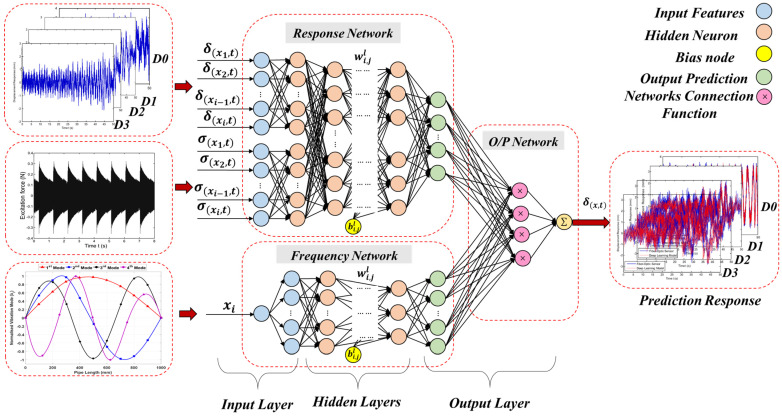
The architecture of the proposed ECNN for the response prediction.

**Figure 18 sensors-23-03887-f018:**
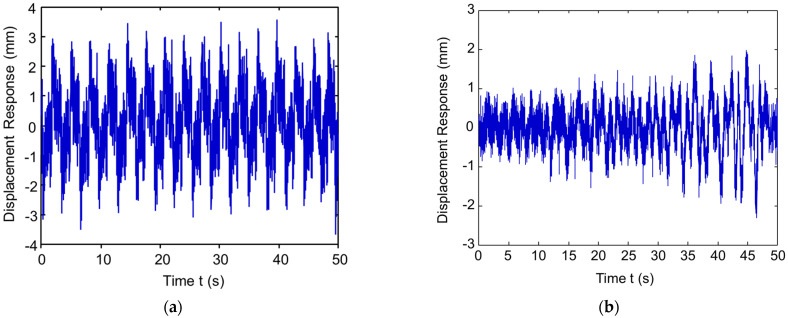
The displacement response of the UDP/DP: (**a**) UDP, D0; (**b**) DP, D1; (**c**) DP, D2; and (**d**) DP, D3.

**Figure 19 sensors-23-03887-f019:**
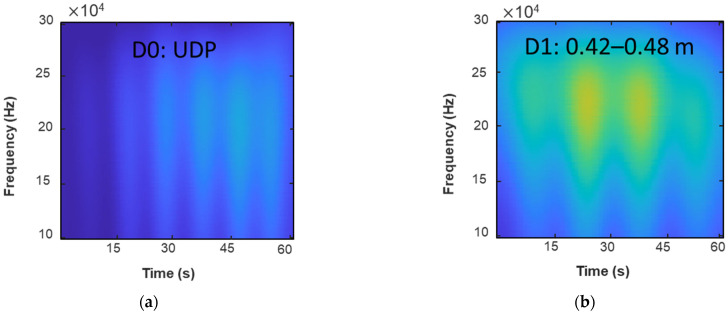
The time–frequency spectrogram (TFS) of the UDP/DP: (**a**) UDP, D0; (**b**) DP, D1; (**c**) DP, D2; and (**d**) DP, D3.

**Figure 20 sensors-23-03887-f020:**
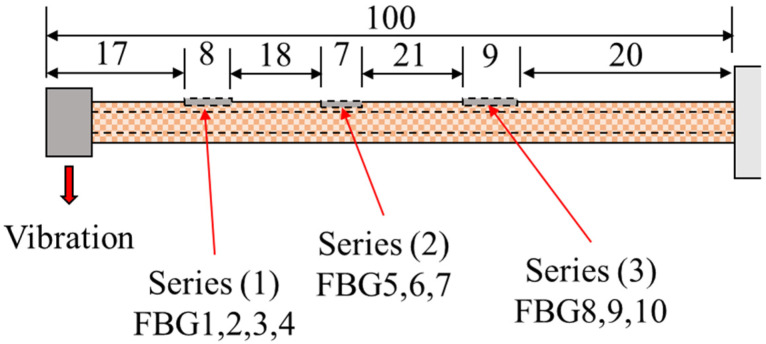
The position of FBGs in the series setup of the pipe.

**Figure 21 sensors-23-03887-f021:**
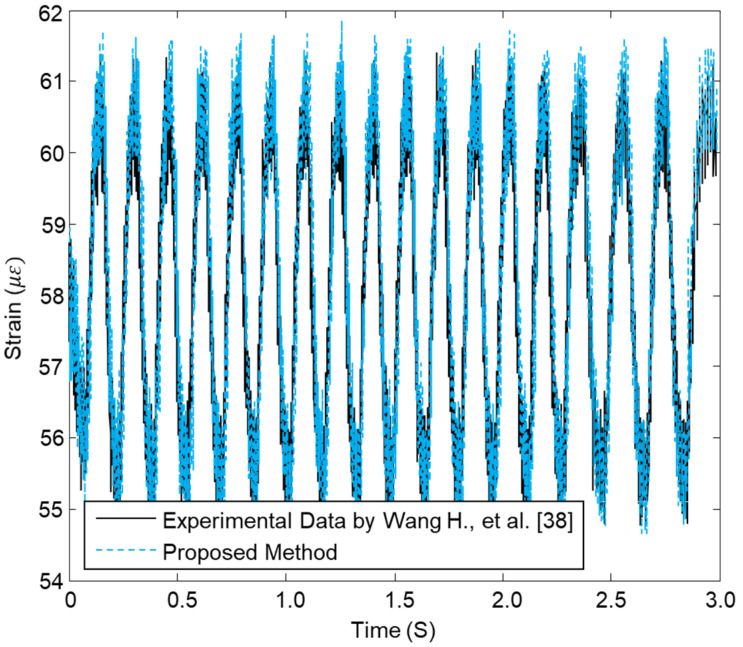
Comparison between the proposed method and experimental values of strain at 10 Hz for FBG2 in a CFRP composite pipe.

**Figure 22 sensors-23-03887-f022:**
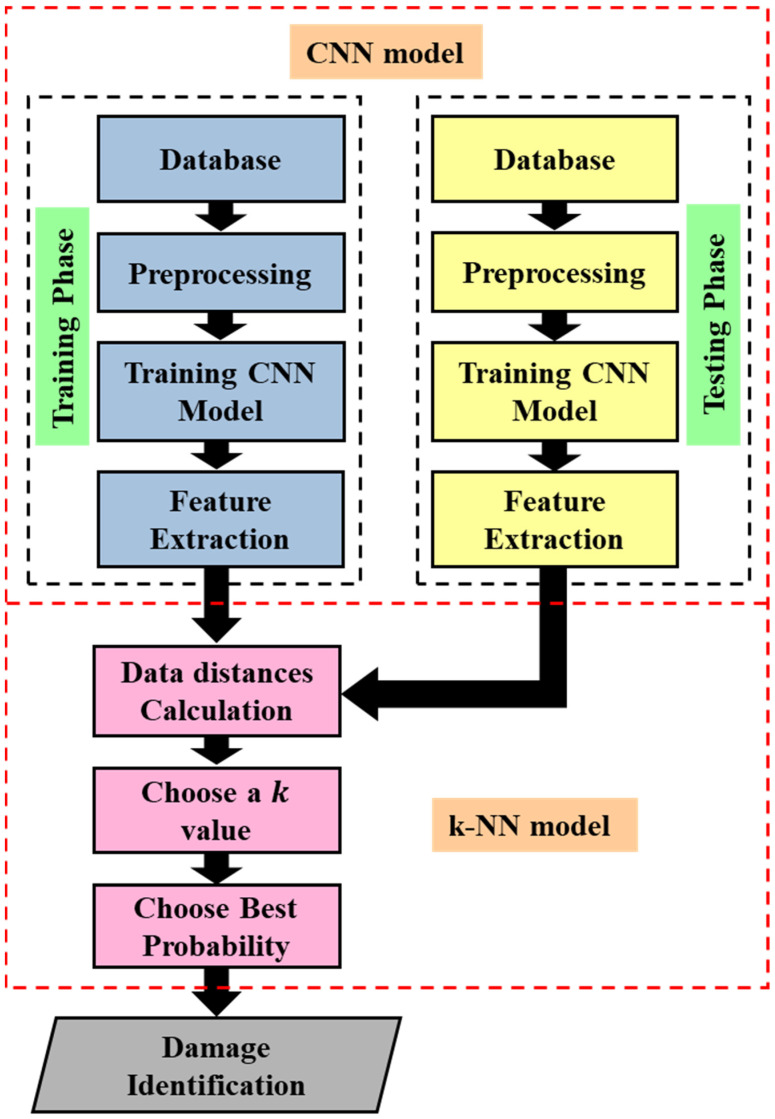
The Overall CNN + k-NN (ECNN) Flowchart.

**Figure 23 sensors-23-03887-f023:**
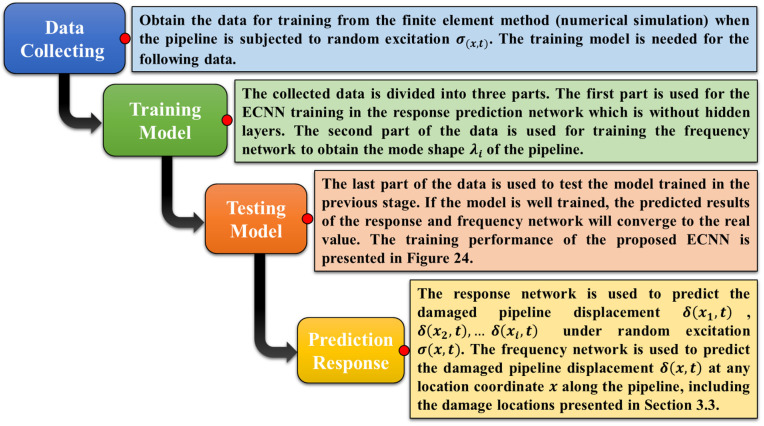
The steps of ECNN development.

**Figure 24 sensors-23-03887-f024:**
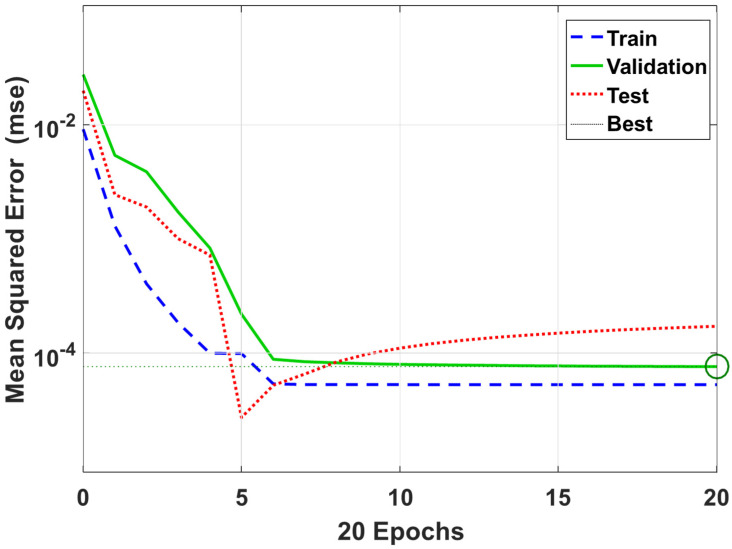
Training performance of proposed ECNN.

**Figure 25 sensors-23-03887-f025:**
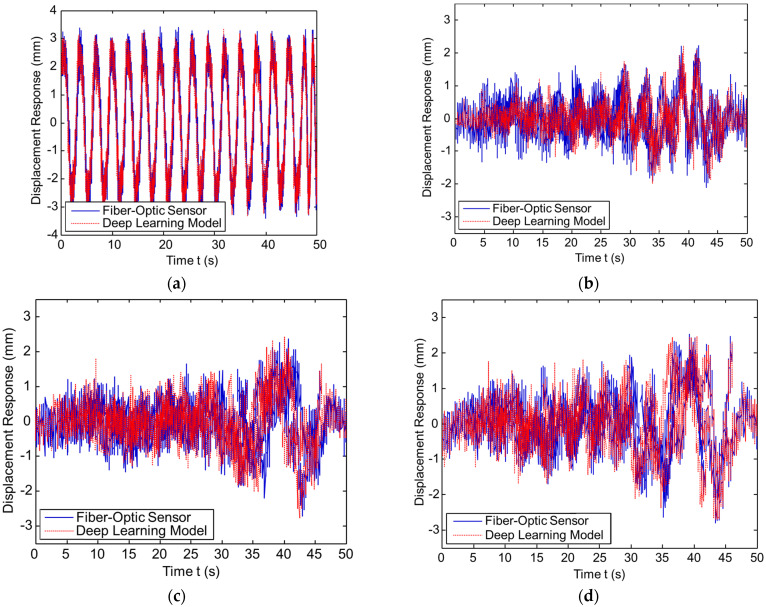
The Displacement response prediction of the UDP and DP under ambient excitation load σ(x,t). (**a**) Response prediction of UDP, D0. (**b**) Response prediction of DP, D1. (**c**) Response prediction of DP, D2. (**d**) Response prediction of DP, D3.

**Figure 26 sensors-23-03887-f026:**
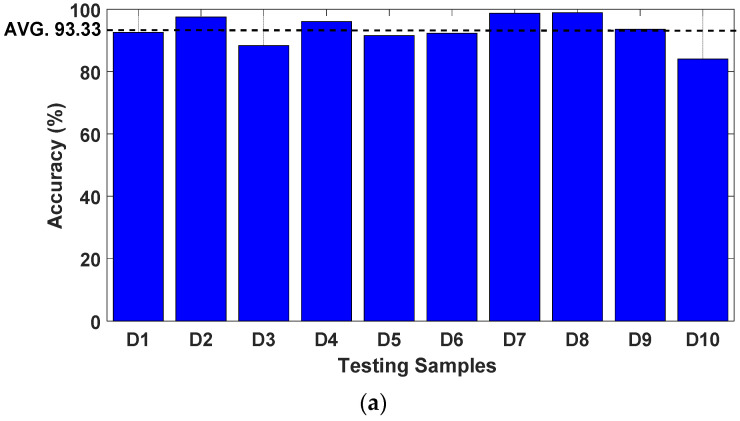
The training process comparison based on the damage identification test data for ten data sets of displacement. (**a**) Accuracy (P%). (**b**) Regression rate (R%). (**c**) F1-score (F%).

**Table 1 sensors-23-03887-t001:** BFRP composite structural properties.

$E_{x}$ $\left(\mathrm{Pa}\times 10{\;}^{9}\right)$	93.5
$E_{y}$ $\left(\mathrm{Pa}\times 10{\;}^{9}\right)$	20
$E_{z}$ $\left(\mathrm{Pa}\times 10{\;}^{9}\right)$	20
$G_{xy}\left(\mathrm{Pa}\times 10{\;}^{9}\right)$	8.5
$G_{yz}$ $\left(\mathrm{Pa}\times 10{\;}^{9}\right)$	2.35
$G_{xz}\left(\mathrm{Pa}\times 10{\;}^{9}\right)$	2.35
$\nu_{xy}$	0.28
$\nu_{yz}$	0.30
$\nu_{xz}$	0.28

**Table 2 sensors-23-03887-t002:** Pipeline frequency orders.

Damage	Frequency Order			
	1st Order	2nd Order	3rd Order	4th Order
D0 (Hz)	233.55	315.62	824.38	3153.85
D1 (Hz)	232.85	312.67	822.90	3151.45
D2 (Hz)	232.13	312.64	822.48	3150.98
D3 (Hz)	230.79	310.11	820.85	3147.73

Remark: D0 is UDP.

**Table 3 sensors-23-03887-t003:** Fiber-optic parameters.

Parameter	Optical Fiber
Elastic Modulus (MPa)	$E_{B}=70$
Diameter (mm)	$D_{B}=0.125$
Area (mm^2^)	$A_{B}=0.01227$
Pitch (m)	$q_{B}=0.1$
Length (m)	$L_{B}=0.86\pi$
Arbitrary orientation	φ=45°

**Table 4 sensors-23-03887-t004:** k-NN internal parameters.

Parameters	Value
Optimum Neighbors number (k)	100
Optimization method	Genetic algorithm (GA)
Distance	Euclidean
Bucket size	50
Include ties	0
Distance weight	equal
Break ties	smallest
Standardize data	1
Type	Prediction
min(X)	[2.840,41.322]
StdDev (X)	[0.485,1.016]
Weight (W)	74.9 × 10^−5^

**Table 5 sensors-23-03887-t005:** Labeled Dataset.

**Label**	1	2	3	4
**Case**	D0	D1	D2	D3
**Location**	UDP	0.42–0.48 m	0.52–0.58 m	0.62–0.68 m

**Table 6 sensors-23-03887-t006:** Identification Testing Detailed Results.

Indexes	AI Method	Label			
		1	2	3	4
TPR	k-NN	97.11%	90.89%	91.73%	93.97%
	TCNN	98%	92%	93%	95.2%
	ECNN	**98.71%**	**95.64%**	**96.32%**	**97.19%**
TNR	k-NN	94%	96.82%	97.13%	95.81%
	TCNN	95.66%	98.53%	97.95%	96.74%
	ECNN	**97.61%**	**99.32%**	**99.46%**	**98.72%**
FPR	k-NN	6.21%	2.14%	4.34%	6.65%
	TCNN	6.35%	2.45%	4.56%	6.99%
	ECNN	**7.59%**	**3.49%**	**6.1%**	**8.32%**
FNR	k-NN	3.87%	11.98%	11.27%	10.34%
	TCNN	4%	13%	12%	11.3%
	ECNN	**6.21%**	**15.4%**	**13.64%**	**12.33%**

## Data Availability

This research received no Data Availability Statement.

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
