# Peer review of "Structural Health Monitoring of Composite Pipelines Utilizing Fiber Optic Sensors and an AI-Based Algorithm—A Comprehensive Numerical Study"

_sensors, 2023, doi:10.3390/s23083887_

Round 1

Reviewer 1 Report

The paper is very well-written. The reviewer supports the acceptance of the manuscript (in present form). The manuscript fits the journal very well. One suggestion that may further improve the quality of the manuscript is to pay attention to the readability of some figures. For instance,  

Figs. 4, 21, and 25: the font can be larger.  

Fig. 24: combined with the different colours, different line types are also suggested to mark the differences. For instance, the ‘Validation’ can be a solid line; the ‘Train’ can be a dashed line; the ‘Test’ is marked with a dash-dot line. It may help readers who prefer to print papers in white and black.  

Author Response

Title: Structural Health Monitoring of Composite Pipelines Utilizing Fiber Optic Sensors and AI-Based Algorithm – A Comprehensive Numerical Study.

Reviewer #1

The paper is very well-written. The reviewer supports the acceptance of the manuscript (in present form). The manuscript fits the journal very well. One suggestion that may further improve the quality of the manuscript is to pay attention to the readability of some figures. For instance, 

Response: We appreciate your positive feedback.

  • 4, 21, and 25: the font can be larger.

Response: We appreciate your positive feedback. We changed Figure 4 with another easier one, and Figures 21, 25 we replaced the font with larger.

  • 24: combined with the different colours, different line types are also suggested to mark the differences. For instance, the ‘Validation’ can be a solid line; the ‘Train’ can be a dashed line; the ‘Test’ is marked with a dash-dot line. It may help readers who prefer to print papers in white and black.

Response: We appreciate your positive feedback. We changed the figure 24 lines as you recommended.

Reviewer 2 Report

1. The technical contents of this paper is quite good and written comprehensively. But the organization of this paper did not follow the common outlines of Section 1 - Introduction, Section 2 - Literature Review, Section 3 -  Methodology, Section 4 - Results and Discussion and Section 5 - Conclusion. I observe that most information in different sections are mixed around and it is very difficult to follow the contents. For instance, Section 3.5.1 A K-NN algorithm and Section 3.5.2 The CNN modeling should be explained under literature review instead of under the Section 3. Results and Discussions. Section 3.5.3 Hybrid CNN + KNN should be explained under methodology. There are too many similar issues and I am not able to list them out exhaustively. But I hope that the authors get my point and make necessary revision.

2. Abstract is quite lengthy. Please make it shorter and more concise. 

3. Section 1.5 Contribution and Section 1.5.1 Methodology can be merged. 

4. Table 1: What are those figures such as 93.5E9, 20E9 and etc. referring to?

5. The meaning of some mathematical symbols such as those in Eqs. (3) and (4) are not defined. Perhaps the authors can consider to add a nomenclature table to summarize all variables and parameters used.

6. Eqs. (30) to (33) are not necessary and can be removed.

7. Eq. (36) should  be known as F1 score instead of F score.

8. A more in-depth discussion is needed to justify the performance gain achieved by ECNN. Authors should provide more critical discussion on how the kNN layer is helpful to achieve performance gain.

9. The proposed method should be compared with state-of-art methods to verify the strengths.

Author Response

Reviewer #2

  • The technical contents of this paper is quite good and written comprehensively. But the organization of this paper did not follow the common outlines of Section 1 - Introduction, Section 2 - Literature Review, Section 3 - Methodology, Section 4 - Results and Discussion and Section 5 - Conclusion. I observe that most information in different sections are mixed around and it is very difficult to follow the contents. For instance, Section 3.5.1 A K-NN algorithm and Section 3.5.2 The CNN modeling should be explained under literature review instead of under the Section 3. Results and Discussions. Section 3.5.3 Hybrid CNN + KNN should be explained under methodology. There are too many similar issues and I am not able to list them out exhaustively. But I hope that the authors get my point and make necessary revision.

Response: We appreciate your positive feedback. We rearranged the sections and now the organization of paper section is not difficult and can be read easily.

  • Abstract is quite lengthy. Please make it shorter and more concise.

Response: We appreciate your positive feedback. We reduced the abstract length and it is now more concise.

  • Section 1.5 Contribution and Section 1.5.1 Methodology can be merged.

Response: We appreciate your positive feedback. We have dedicated a section on methodology and restructured the paper in general.

  • Table 1: What are those figures such as 93.5E9, 20E9 and etc. referring to?

Response: We appreciate your positive feedback. This is the values of mechanical properties of composite material used in paper, where , ,  are elastic modulus in the ‘x’, ‘y’ and ‘z’ directions, , ,  are shear modulus in the ‘xy’, ‘xz’ and ‘yz’ plane, and , ,  are Poisson's ratio in the ‘xy’, ‘xz’ and ‘yz’ plane. We have written these in the manuscript before  Table 1.

  • The meaning of some mathematical symbols such as those in Eqs. (3) and (4) are not defined. Perhaps the authors can consider to add a nomenclature table to summarize all variables and parameters used.

Response: We appreciate your positive feedback. We defined all  symbols in all equations in the paper, now all symbols definitions are provided in equations or in Figures. The nomenclature table is placed in the end of the paper for all abbreviations defined in the paper.

  • (30) to (33) are not necessary and can be removed.

Response: We appreciate your positive feedback. We removed (6). Eqs. (30) to (33).

  • (36) should be known as F1 score instead of F score.

Response: We appreciate your positive feedback. We corrected that.

  • A more in-depth discussion is needed to justify the performance gain achieved by ECNN. Authors should provide more critical discussion on how the kNN layer is helpful to achieve performance gain.

Response: We appreciate your positive feedback. We have thoroughly disscused the performance gain achieved by ECNN and compared with ECNN  performance gain and other confegurations used in this paper and you can see that clearly in Table 6 and Figure 26. We followed the majority of methods used in the literature for assesment of the performance gain for the three configurations of deep learning and machine learning.

  • The proposed method should be compared with state-of-art methods to verify the strengths.

Response: We appreciate your positive feedback and thank you for this great suggestion. In this work the results of the proposed method are already compared with experimental results available in the literature and also compared with predicted results from three configurations of deep learning and machine learning, thus validating the accuracy and reliability and verifying the strengths of the proposed technique.

Reviewer 3 Report

The paper

“Structural Health Monitoring of Composite Pipelines Utilizing Fiber Optic Sensors and AI-Based Algorithm – A Comprehensive Numerical Study”,

By Altabey et al.,

Presents an early warning damage detection system, based on readings from an embedded Fibre Bragg grating (FBG) sensory system and intended for applications on basalt fibre reinforced polymer (BFRP) composite pipelines. The procedure applies Enhanced Convolutional Neural Network (ECNN) and k-Nearest Neighbor (k-NN) algorithm to such measurements.

The content of the manuscript is in line with the aim of the journal and the S.I. and can be considered for potential publication.

Nevertheless, this Reviewer would like to highlight the following aspects which can be further improved to make the article even more compelling for researchers and practitioners operating in the fields of SHM:

1.      With circa 400 words, the abstract is too long and should be shortened.

2.       The flowchart in figure 4 is very large, complex, and a bit convoluted with arrows splitting and pointing in different directions. The use of many writings with small font sizes and many plots with many details and legends also does not help. In the end, the final result is very difficult to follow. It is strongly suggested to simplify the flowchart and/or split it into multiple subsections (each one with its own flowchart).

3.      The structure of the manuscript should be reconsidered. Generally, the ‘Methodology’ portion is a fully-fledged Section on its own, not a sub-sub section of the Introduction. The FE model of the pipeline should be reported in a dedicated section as well rather than in the Results.

4.       The subsection of the Introduction lists four papers, all from the past decade (the newest one is from 2009). There is a large scientific literature about Damage Detection and Localisation in Buried Pipelines, also belonging to the same journal (Applied Sciences). Some of the most recent contributions in the field can be added to provide more context to the research reported here.

5.      Also related to the previous remark, it is not clear why some other references are then listed in separate state-of-the-art reviews in Sec 1.3 and Sec 1.4.1. Since there is a dedicated section, it would be better to include all of them in the same part of the manuscript.

6.      Sect. 2.1: This reviewer does not think that the use of bold characters in the main text is needed or advisable.

7.       Table 1 and elsewhere: the writing 93.5 10^9 is generally more appreciated than 93.5E9. Also, the material properties clearly follow the Ansys nomenclature convention, which is easy to understand for any Ansys user. Yet, for readers not experienced with that specific software (also for good practise in scientific writing overall), all acronyms should be explained (EX: Young’s Modulus in the x direction, etc). Finally, if the x-y and x-z Poisson’s Ratios are reported with two decimal digits, also y-z should be 0.30

8.      Is figure 5 an original contribution to this paper or is it retrieved from the existing scientific literature? In the second case, the source should be added in the figure caption. The same consideration applies to other figures and sub-figures as well (e.g. Fig. 8.a).

9.      It is not totally clear to this reviewer if the input signal (the ambient excitation force shown in Figure 11) is experimentally recorded or numerically simulated.

10.   The caption of Figure 12 states  "The First three frequency orders of an intact pipeline." However, the figure seems to show four modes. Plus, mode shapes, not frequency orders, are shown. Also, the mode shapes should be normalised to be consistent with the boundary conditions (as they seem to all begin and end at nonzero values).

11.   The displacement responses reported in Figures 13 and 25 are all on the order of magnitude of +-1 mm, which seems to be a bit high for ambient vibrations.

12.   Table 4: why is Euclidean the only term in italics?

13.   Fig. 22 is much larger than the previous Fig. 21 and should be resized. Generally speaking, for consistency, the font size of writings inside figures should be more or less the same as the font size of the figure caption.

14.   At the end of the manuscript, the 'Funding' and 'Data Availability Statement' need some corrections.

Author Response

Reviewer #3

Presents an early warning damage detection system, based on readings from an embedded Fibre Bragg grating (FBG) sensory system and intended for applications on basalt fibre reinforced polymer (BFRP) composite pipelines. The procedure applies Enhanced Convolutional Neural Network (ECNN) and k-Nearest Neighbor (k-NN) algorithm to such measurements.

The content of the manuscript is in line with the aim of the journal and the S.I. and can be considered for potential publication.

Nevertheless, this Reviewer would like to highlight the following aspects which can be further improved to make the article even more compelling for researchers and practitioners operating in the fields of SHM:

  • With circa 400 words, the abstract is too long and should be shortened.

Response: We appreciate your positive feedback. We reduced the abstract length and it is now more concise, the abstract is now the standard length of 300 words.

  • The flowchart in figure 4 is very large, complex, and a bit convoluted with arrows splitting and pointing in different directions. The use of many writings with small font sizes and many plots with many details and legends also does not help. In the end, the final result is very difficult to follow. It is strongly suggested to simplify the flowchart and/or split it into multiple subsections (each one with its own flowchart).

Response: We appreciate your positive feedback. We simplified the flowchart Figure 4 and now it is easier to read.

  • The structure of the manuscript should be reconsidered. Generally, the ‘Methodology’ portion is a fully-fledged Section on its own, not a sub-sub section of the Introduction. The FE model of the pipeline should be reported in a dedicated section as well rather than in the Results.

Response: We appreciate your positive feedback. We have dedicated a section on methodology and FE model and restructured the paper in general.

  • The subsection of the Introduction lists four papers, all from the past decade (the newest one is from 2009). There is a large scientific literature about Damage Detection and Localisation in Buried Pipelines, also belonging to the same journal (Applied Sciences). Some of the most recent contributions in the field can be added to provide more context to the research reported here.

Response: We appreciate your positive feedback. We added more sentences in subsection of the Introduction and added more references in 2022, 2023.

  • Also related to the previous remark, it is not clear why some other references are then listed in separate state-of-the-art reviews in Sec 1.3 and Sec 1.4.1. Since there is a dedicated section, it would be better to include all of them in the same part of the manuscript.

Response: We appreciate your positive feedback. We transfered all subsections from introduction to the dedicated sections to include all of them in the same part of the manuscript as you recommended.

  • 2.1: This reviewer does not think that the use of bold characters in the main text is needed or advisable.

Response: We appreciate your positive feedback. We removed all bold characters in the main text.

  • Table 1 and elsewhere: the writing 93.5 10^9 is generally more appreciated than 93.5E9. Also, the material properties clearly follow the Ansys nomenclature convention, which is easy to understand for any Ansys user. Yet, for readers not experienced with that specific software (also for good practise in scientific writing overall), all acronyms should be explained (EX: Young’s Modulus in the x direction, etc). Finally, if the x-y and x-z Poisson’s Ratios are reported with two decimal digits, also y-z should be 0.30

Response: We appreciate your positive feedback. We  have written all acronyms in the paper text before Table 1, also we corrected the x-z Poisson’s Ratios in two decimal digits, and we have written 93.5 10^9  instead of 93.5E9 in Table 1.

  • Is figure 5 an original contribution to this paper or is it retrieved from the existing scientific literature? In the second case, the source should be added in the figure caption. The same consideration applies to other figures and sub-figures as well (e.g. Fig. 8.a).

Response: We appreciate your positive feedback. Figure 5 is an original contribution to this paper, and Fig. 8.a, now is also an original contribution and we confirm that all other figures in this paper are original contributions.

  • It is not totally clear to this reviewer if the input signal (the ambient excitation force shown in Figure 11) is experimentally recorded or numerically simulated.

Response: We appreciate your positive feedback. The input signal in Figure 11 is numerically simulated, and we added this notes in paper text.

  • The caption of Figure 12 states "The First three frequency orders of an intact pipeline." However, the figure seems to show four modes. Plus, mode shapes, not frequency orders, are shown. Also, the mode shapes should be normalised to be consistent with the boundary conditions (as they seem to all begin and end at nonzero values).

Response: We appreciate your positive feedback. We corrected all your comments thanks a lot.

  • The displacement responses reported in Figures 13 and 25 are all on the order of magnitude of +-1 mm, which seems to be a bit high for ambient vibrations.

Response: We appreciate your positive feedback. This is due to the phenomena of the stiffness of composite materials especially BFRP is softer than other types of FRP. We have checked all results in Figures 13, 25 and foud them to be correct.

  • Table 4: why is Euclidean the only term in italics?

Response: We appreciate your positive feedback. We corrected this error.

  • 22 is much larger than the previous Fig. 21 and should be resized. Generally speaking, for consistency, the font size of writings inside figures should be more or less the same as the font size of the figure caption.

Response: We appreciate your positive feedback. In general we modified the font size in all Figures specially Figure 21.

  • At the end of the manuscript, the 'Funding' and 'Data Availability Statement' need some corrections.

Response: We appreciate your positive feedback. We added the funding source of  project of this paper, and other available data in the end of this paper.

Round 2

Reviewer 2 Report

The quality of the manuscript has been improved significantly after the revisions. Thanks to the authors for putting serious efforts to close all concerns raised. The quality of revised manuscript is good enough to be accepted for publication. 

Author Response

We believe this had been addressed already.

Reviewer 3 Report

The authors provided several ameliorations to their manuscript in the new, revised version. Content-wise, the main problems have been positively addressed. However, some issues (mainly editorial ones) raised by this Reviewer during the first round of review still need to be better addressed. Specifically:

This Reviewer is satisfied with the replies and modifications made regarding comments 1, 3, 5, 6, 8, 9, 10, 11, and 12.

Regarding comment 2:

The flowchart in figure 4 is still difficult to read. The Authors did simplify the flowchart considerably. Yet, even in its current form, many figures are complex and with many small writings included in them.

Regarding comment 4:

The state of the art is still relatively limited and scattered throughout the manuscript. A suggested addition to the discussion is reported at https://doi.org/10.3390/app11135773.

Regarding comment 7:

This Reviewer meant 109 (ten superscripts 9), not literally 10^9

Regarding comment 13:

Overall, the quality of the figures can still be improved (e.g. the font size in the x and y labels of Figure 18 are still very small and thus difficult to read).

Also, there are still many typos and grammar errors, both in the previous text and the newly-added sections. For instance, in the legend of Figure 5, The [f]irst [f]our mode shapes of an intact pipeline.

Author Response

Title: Structural Health Monitoring of Composite Pipelines Utilizing Fiber Optic Sensors and AI-Based Algorithm – A Comprehensive Numerical Study.

Reviewer #3

The authors provided several ameliorations to their manuscript in the new, revised version. Content-wise, the main problems have been positively addressed. However, some issues (mainly editorial ones) raised by this Reviewer during the first round of review still need to be better addressed. Specifically:

This Reviewer is satisfied with the replies and modifications made regarding comments 1, 3, 5, 6, 8, 9, 10, 11, and 12.

  • Regarding comment 2:

The flowchart in figure 4 is still difficult to read. The Authors did simplify the flowchart considerably. Yet, even in its current form, many figures are complex and with many small writings included in them.

Response: We appreciate your positive feedback. We modified the flowchart more easier and all writings included in them are clear

  • Regarding comment 4:

The state of the art is still relatively limited and scattered throughout the manuscript. A suggested addition to the discussion is reported at https://doi.org/10.3390/app11135773.

Response: We appreciate your positive feedback. We discussed about this reference in introduction and added this paper in references.

  • Regarding comment 7:

This Reviewer meant 109 (ten superscripts 9), not literally 10^9

Response: We appreciate your positive feedback. We corrected this mistake.

  • Regarding comment 13:

Overall, the quality of the figures can still be improved (e.g. the font size in the x and y labels of Figure 18 are still very small and thus difficult to read).

Response: We appreciate your positive feedback. We modified the font size in the x and y labels in Figure 18 and Figure 25.
